# Learning to Solve SMT Formulas

**Mislav Balunović, Pavol Bielik, Martin Vechev**
Department of Computer Science
ETH Zürich
Switzerland
bmislav@ethz.ch, {pavol.bielik, martin.vechev}@inf.ethz.ch

## Abstract

We present a new approach for learning to solve SMT formulas. We phrase the challenge of solving SMT formulas as a tree search problem where at each step a transformation is applied to the input formula until the formula is solved. Our approach works in two phases: first, given a dataset of unsolved formulas we learn a policy that for each formula selects a suitable transformation to apply at each step in order to solve the formula, and second, we synthesize a *strategy* in the form of a loop-free program with branches. This strategy is an interpretable representation of the policy decisions and is used to guide the SMT solver to decide formulas more efficiently, without requiring any modification to the solver itself and without needing to evaluate the learned policy at inference time. We show that our approach is effective in practice – it solves 17% more formulas over a range of benchmarks and achieves up to 100× runtime improvement over a state-of-the-art SMT solver.

## 1 Introduction

Satisfiability Modulo Theories (SMT) solvers are a powerful class of automated theorem provers which can deduce satisfiability and validity of first-order formulas in particular logical theories (e.g., real numbers, arrays, bit vectors). SMT solvers are more general than SAT solvers and have been used in a variety of application domains including verification (e.g., neural networks [28]), program synthesis, static analysis, scheduling, and others [16].

To efficiently solve complex real world problems, state-of-the-art SMT solvers (e.g., Z3 [15], Yices [18], CVC4 [4], MathSAT5 [13], Boolector [35]) contain hand-crafted heuristics combining algorithmic proof methods and satisfiability search techniques. Indeed, crafting the right heuristic is critical and can be the difference between solving a complex formula in seconds or not at all. To enable end users to create suitable heuristics, several modern SMT solvers such as Z3 [15] provide a so called tactic language for expressing heuristics. Typically, such tactics are combined and performed in a sequence (specified by the user), forming an interpretable program called a *strategy*. However, the resulting strategies can often end up being quite complex (e.g., containing loops and conditionals). Combined with the vast search space, this means that manually finding a well-performing strategy can be very difficult, even for experts.

**Our Contributions**   We present a new approach, based on a combination of learning and synthesis techniques, which addresses the problem of automatically finding the right SMT strategy. Given a dataset of formulas whose solution is initially unknown, we first train a policy that searches for strategies that are fast at solving the formulas in this dataset. Then, we synthesize a single strategy (a program with branches) that captures the policy decision making in an interpretable manner. The resulting strategy is then passed on to the SMT solver which uses it to make more effective decisions when solving a given formula. We note that our approach does not require any changes to the solver's internals and thus can work with any decision procedure which cleanly exports its tactics.

We performed an extensive experimental evaluation of our approach on formulas of varying complexity across 3 different theories (`QF_NRA`, `QF_BV` and `QF_NIA`). We show that our synthesized strategies solve $17\%$ more formulas and are up to $100\times$ faster when compared to the default strategy used in the state-of-the-art Z3 solver. Further, our learned strategies generalize well and can solve formulas which are much more challenging than those seen during training. Finally, we make our tool, called `fastSMT`, datasets and experiments available at `http://fastsmt.ethz.ch/`.

## 2 Related work

Given the importance and wide range of SMT solver applications, a number of approaches have been developed to improve both, their runtime as well as the range of formulas they can solve.

**Portfolio based approaches**   The most common approach of tools such as `SATzilla` [44], `CPhydra` [12], `SUNNY` [2], `Proteus` [22], `ISAC` [27] is a portfolio based method. The key idea is that different SMT solvers use different heuristics and hence work well for different types of formulas. Then, given several SMT solvers and a dataset of formulas, the goal is to learn a classifier which uses features extracted from the given formula to select the right SMT solver (or alternatively defines order in which to run the solvers). In comparison, we address a harder problem - learn how to instantiate an SMT solver with a strategy that efficiently solves the given dataset of formulas.

**Evolutionary search**   The work most closely related to ours is `StratEVO` [39] which also studies the task of generating SMT strategies. However, `StratEVO` has several limitations – it performs search using an evolutionary algorithm which does not incorporate any form of learning, the search does not depend on the actual formula, and local mutations tend to get stuck in local minima (as we show in our experiments in Section 5). As a result, for many tasks where the suitable strategy cannot be trivially found (e.g., is very short) their approach reduces to random search. Instead, we leverage models that learn from previously explored strategies as well as the current formula. As we show, this enables discovery of complex strategies that are out of reach for approaches not based on learning.

**Learning branching heuristics and premise selection**   Recently, learning techniques have been applied to improving performance of SAT solvers [32, 42], constraint programming solvers [34], solving quantified boolean formulas [41], solving mixed integer programming problems [29] as well as premise selection in interactive theorem provers [1, 43, 33]. At a high level, these are complementary to us – we learn to search across many tactics and combine them into high level strategies while they optimize a single tactic (e.g., by learning which variable to branch on in SAT). Our work also supports formulas from a wide range of theories (as long as there is a corresponding tactic language) where selecting a suitable high level strategy leads to higher speedups compared to optimizing a single tactic in isolation. However, there are also common challenges such as defining a suitable formula representation. This representation can range from a set of hand-crafted features [32, 36], to recursive and convolutional neural networks [33, 1], to graph embeddings [43]. We extend this line of work by considering fast to compute representations based on bag of words, syntactic features, and features extracted from a graph representation of the formula.

**Parameter tuning**   Finally, a number of approaches exist for finding good parameter configurations from a vast space of both discrete and continuous parameters, including `ParamILS` [25], `GGA` [3], $TB-SPO$ [24] or `SMAC` [23]. An interesting application of such approaches to the domain of SAT/SMT solvers is proposed by `SATenstein` [30] which first designs a highly parameterized solver framework to be subsequently optimized by off-the shelf tools. Although such tools are not applicable for the task of searching for strategy programs (that include loops and conditionals) considered in our work, they can be used to fine-tune the strategy parameters once a candidate strategy is found.

## 3 SMT strategies: preliminaries

At a high level, an SMT solver takes as input a first-order logical formula and then tries to decide if the formula is satisfiable. In the process, solvers employ various heuristics that first transform the input formula into a suitable representation and then use search procedures to check for satisfiability. Existing heuristics include: `simplify` which applies basic simplification rules such as constant

Table 1: Formalization of the `Strategy` language used to express SMT strategies in Z3 [15].

| (Strategy) | q | ::= | `t | q; q | if p then q else q | q else q |` |
| | | | `repeat q, c | try q for c | using t with params` |
| (Tactics) | t | $\in$ | Tactics = { `bit_blast, solve_eqs, elim_uncnstr` ... } |
| (Predicates) | p | ::= | $p \wedge p \mid p \vee p \mid$ `expr` $\bowtie$ `expr` |
| (Expressions) | expr | ::= | `c | probe | expr` $\oplus$ `expr` |
| (Constants) | c | $\in$ | Consts = $\mathbb{Q}$ |
| (Probes) | probe | ::= | Probe $\rightarrow \mathbb{Q}$, Probe = { `num_consts, is_pb`, ... } |
| (AOperators) | $\oplus$ | ::= | $+ \mid - \mid * \mid /$ |
| (BOperators) | $\bowtie$ | ::= | $> \mid < \mid \geq \mid \leq \mid = \mid \neq$ |
| (Parameter) | param | ::= | (Param, $\mathbb{Q}$), Param = { `hoist_mul, flat, som`, ... } |
| (Parameters) | params | ::= | $\epsilon \mid$ `param; params` |

folding ($x + 0 \rightarrow x$), removal of redundant expressions ($x - x \rightarrow 0$), `gaussian_elim` which eliminates variables ($x = 1 \wedge y \geq x + z \rightarrow y \geq 1 + z$) using Gaussian elimination, `elim_term_ite` which replaces the term if-then-else with fresh auxiliary declarations ($($ `if` $x > y$ `then` $x$ `else` $y) >$ $z \rightarrow k > z \wedge (x > y \Rightarrow k = x) \wedge (x \leq y \Rightarrow k = y))$, or `bit_blast` which reduces bit-vector expressions by introducing fresh variables, one for each bit of the original bit-vector (e.g., a bit vector of size 4 is expanded into four fresh variables). In total, the Z3 SMT solver defines more than 100 such heuristic transformations (called tactics) that can be combined together to define a custom strategy. For example, a strategy for integer arithmetic can be defined as[1]:

`using simplify with (som: true); normalize_bounds; lia2pb; pb2bv; bit_blast; sat`

Although the above sequence of transformations (tactics) works well for some types of input formulas (e.g., in case every variable has a lower and an upper bound), for other formulas a different set of tactics is more suited. In some cases, the suitable set of tactics can be obtained by a small modification of the original tactic while in others, a completely different set of tactics needs to be defined. As a concrete example, consider the following strategy implemented in the Yices SMT solver [18, 17]:

$$\texttt{if} \left( \neg \texttt{diff} \wedge \frac{\texttt{num\_atoms}}{\texttt{dim}} < k \right) \texttt{then } \texttt{simplex} \texttt{ else } \texttt{floyd\_warshall}$$

Here, two high level tactics can be applied to solve an input formula – the Simplex algorithm or the algorithm based on Floyd-Warshall all-pairs shortest distance. The Simplex algorithm is used if the input formula is not in the difference logic fragment (denoted using $\neg$`diff`) and the ratio of inequalities divided by the number of uninterpreted constants is smaller than a threshold $k$.

**Language for expressing SMT solver strategies** In Fig. 1 we formalize the language used by Z3 to enable control over the core solver heuristics. The basic building blocks of the language are called tactics and represent various heuristic transformations that might be applied during the search. Optionally, each tactic can define a set of parameters that affect its behavior. For example, the `simplify` tactic contains $> 50$ different parameters that control which simplifications are performed (e.g., if `som : true` then `simplify` puts polynomials in som-of-monomials form). The tactics are combined into larger programs either sequentially or using one of the following control structures:

- `if p then` $q_1$ `else` $q_2$: If the predicate p evaluates to `true` apply $q_1$, otherwise apply $q_2$. The predicate can contain arithmetic expressions as well as Probes which collect statistics of the current formula (e.g., `num_consts` returns the number of non-boolean constants).

- $q_1$ `else` $q_2$: First apply $q_1$ and if it fails then apply $q_2$ on the original input.

- `repeat` q, c: Keep applying q until it does not modify the input formula any more or the number of iterations is greater than the constant c.

- `try` q `for` c: Apply q to the input and if it does not return in c milliseconds then fail.

- `using` t `with` params: Apply the given tactic t with the given parameters `params`.

# 4 Learning SMT strategies

We now present our approach for synthesizing SMT strategies. Formally, the task we are interested in solving is defined as follows:

**Problem statement**   Given a dataset of SMT formulas $\mathcal{F} = \{f_i\}_{i=1}^N$, our goal is to find a strategy:

$$q_{best} = \underset{q \in \mathtt{Strategy}}{\arg \min} \sum_{i=1}^N cost(f_i, q) \quad \text{where } cost(f_i, q) \overset{\text{def}}{=} \begin{cases} \mathtt{runtime}(q(f_i)) & \text{if } q \text{ solves } f_i \\ \mathtt{timeout\_penalty} & \text{otherwise} \end{cases} \quad (1)$$

Here, $\mathtt{runtime}(q(f_i)) \in \mathbb{Q}$ denotes the runtime required for strategy $q$ to solve formula $f_i$ and $\mathtt{timeout\_penalty} \in \mathbb{Q}$ is a constant denoting the penalty for not solving $f_i$ (either due to timeout or because the strategy $q$ is not powerful enough). Our goal is to find a strategy that minimizes the time required to solve the formulas in the dataset $\mathcal{F}$. Note that our *cost* function reflects the fact that we aim to synthesize a strategy that solves as many formulas as possible yet one that is also the fastest. Generally, optimizing for Equation 1 directly is problematic as using runtime makes the optimization problem inherently noisy and non-deterministic. It also makes learning hard to parallelize due to significant impact of hardware and environment on overall runtime. Thus, instead of runtime, we use the amount of work measured as the number of basic operations performed by the solver required to solve a formula (e.g., implemented via the `rlimit` counter in Z3).

**Challenges**   To solve the problem of learning SMT strategies, one has to address two challenges:

- *Interpretability*. We are interested in learning a model that is not only efficient at solving a given set of formulas but also expressible as programs in the `Strategy` language. This is important as the learned strategies can then be directly used an input to existing solvers.

- *No prior domain knowledge*. Our learning does not assume any prior knowledge about the dataset $\mathcal{F}$ and which strategies work best in general – initially the solution to all the formulas in the dataset is unknown, no existing strategies are used to boostrap the learning (not even the default strategies already written by the SMT solver developers) and we do not rely on any heuristics (and their combination) that may be useful in solving formulas from $\mathcal{F}$. Indeed, this represents the most challenging setting for learning.

**Our approach**   The key idea behind learning a program $q_{best} \in \mathtt{Strategy}$ efficiently is to take advantage of the fact that each program $q$ can be decomposed into a set of smaller branch-free programs $q_{1,\dots,k}$, each $q_i$ corresponding to one execution path of $q$. This is possible because programs in the `Stategy` language do not contain state (all state is implicitly captured in the formula being solved). As a result, in our approach we learn a program $q_{best} \in \mathtt{Strategy}$ via a two step process:

- *Learn candidate strategies*. First, we learn a policy which finds a set of candidate strategies consisting of only sequences of tactics where each strategy performs well on a different subset of the dataset $\mathcal{F}$. This allows us to phrase the learning problem as a tree search over tactics for which state-of-the-art models can be used. This step is described in Section 4.1.

- *Synthesize a combined strategy*. Then, given a set of candidate strategies, we combine these into a single best strategy $q_{best}$ by synthesizing control structures such as branches and loops as supported by the `Strategy` language. This step is described in Section 4.2.

## 4.1 Learning candidate strategies

We phrase the problem of learning candidate strategies as a tree search problem. We start from an initial state $s_0$ and at each timestep $t$ we choose an action $a_t$ that expands the tree by exploring a single edge. In our setting, a state corresponds to a tuple consisting of an SMT formula and a strategy used to compute the formula. An action is described by a tactic and its parameters (if any) $a \in \mathtt{Tactics} \times \mathtt{Params}$. Applying an action transforms the formula into another one. Terminal states are those that decide the initial formula $f$, that is, $f$ was reduced to a form that is either trivially satisfiable or unsatisfiable. Further, for practical reasons, terminal states also include those to which applying an action (tactic) leads to a timeout.

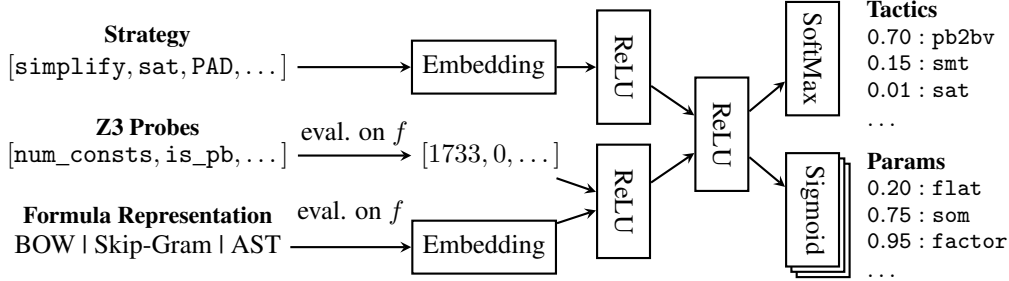

Figure 1: Architecture of the *neural network* model which predicts next tactic and arguments.

A terminal state defines a strategy $q$ (i.e., a sequence of tactics starting from the tree root) with an associated $cost(f, q)$ (as defined in Equation 1) that we aim to minimize. During the search we keep a priority queue of tuples $(s, a, p)$ consisting of a state, possible action and its associated probability, initialized with $(s_0, \epsilon, 1)$. At each step, we remove the tuple with highest probability from the queue, apply its action to obtain a new state $s_i$ and update the priority queue with new tuples $(s_i, a, p \cdot \pi(a \mid s_i))$ for all actions $a \in \texttt{Tactics} \times \texttt{Params}$ capturing the possible transitions from $s_i$. Our goal is to learn a policy, denoted as $\pi(a \mid s)$, that represents a probability distribution of actions conditioned on a state $s$. Ideally, the learned distribution should be such that selecting the most likely action at that state minimizes the expected $cost$. In what follows we first describe the models used to represent the policy $\pi$ considered in our work and then describe how the models are trained, including how to construct a suitable training dataset.

**Bilinear model** Based on matrix factor models used in unsupervised learning of word embeddings [11, 38] as well as supervised text classification [26] we define a bilinear model as follows:

$$\pi(a \mid s) = \sigma(\boldsymbol{U}\boldsymbol{V}\phi(s, a))$$

where $\phi(s, a)$ is a bag of features computed for action $a$ taken in state $s$, $\boldsymbol{U} \in \mathbb{R}^{k \times h}$ and $\boldsymbol{V} \in \mathbb{R}^{h \times |V|}$ are the learnable weight matrices, $V$ is the input vocabulary and $\sigma$ is softmax computing the probability distribution over output classes (in our case actions to be taken in a given state $s$). As the set of all possible actions is too large to consider, we randomly generate a subset of parameters for each tactic before training starts, thus obtaining the overall set of actions $S \subseteq \texttt{Tactics} \times \texttt{Params}$. We define $\phi(s, a)$ to be all n-grams constructed from the strategy of state $s$. For example, for strategy $a_1; a_2; a_3$, the extracted n-grams (features) are described by the vector: $\langle a_1, a_2, a_3, a_1 a_2, a_2 a_3, a_1 a_2 a_3 \rangle$. Then, the vocabulary set $V$ is simply the collection of all possible n-grams formed over $S$.

**Neural network** Our second model is based on a neural network and improves over the bilinear model in two key aspects – it considers a richer state when making the predictions and predicts tactics as well as their corresponding parameters. The architecture of the neural network model is illustrated in Fig. 1 and uses two inputs: (i) the strategy in the current state (as in the bilinear model), and (ii) the formula $f$ to be solved in the current state. The strategy is encoded by first padding it to be of fixed size, concatenating the embedding of each action in the strategy into a single vector and finally feeding the resulting vector into a fully-connected layer. We encode the input formula in two ways: first, by computing a set of formula measures such as the number of expressions and constants, and second, by learning a representation of the formula itself. The formula measures are computed using probes supported by Z3 and are a subset of the possibilities one could define. For the learned representation of the formula we experimented with three different approaches:

- **Bag-of-words (BOW)**: The formula is treated as a sequence of tokens from the SMT-LIB language. We obtain its bag-of-words and use it as the formula embedding.

- **Abstract Syntax Tree (AST)**: From the formulas's AST, we extract all subtrees of depth at most two. The bag-of-words over such subtrees is used as the formula embedding. Note that subtrees which appear in less than 5% of formulas in $\mathcal{F}$ are discarded.

- **Skip-gram**: Each formula in the dataset is treated as a sequence of tokens over which we learn a skip-gram model. We define embedding of the formula as the average of all embeddings of its tokens.

---
**Algorithm 1:** Iterative algorithm used to train policy $\pi$

---
**Data:** Formulas $\mathcal{F}$, Number of iterations $N$, Number of formulas to sample $K$, Exploration rates $\boldsymbol{\beta}$, Exploration policy $\pi_{explore}$ (e.g., random policy)
**Result:** Trained policy $\pi$, Explored strategies $\mathcal{Q}$

1   $\mathcal{D} \leftarrow \emptyset;$    $\mathcal{Q} \leftarrow \emptyset;$    $\pi \leftarrow$ policy initialization
2   **for** $i = 1$ **to** $N$ **do**
3     $\hat{\pi} \leftarrow \boldsymbol{\beta}_i \pi + (1 - \boldsymbol{\beta}_i)\pi_{explore}$           $\triangleright$ policy $\hat{\pi}$ explores with probability $(1 - \boldsymbol{\beta}_i)$
4     $\mathcal{Q} \leftarrow \mathcal{Q} \cup$ Top $K$ most likely strategies for each formula in $\mathcal{F}$ according to $\hat{\pi}$
5     $\mathcal{D} \leftarrow \mathcal{D} \cup$ Extract training dataset from strategies $\mathcal{Q}$
6     $\pi \leftarrow$ Retrain model $\pi$ on $\mathcal{D}$

---

The network output consists of two parts – a probability distribution over tactics to apply next and an assignment to each tactic parameter. The possible tactic parameters are captured by the set `Param` from Fig. 1. We provide a full list of tactics and their parameters in the extended version of our paper. To compute arguments for the tactic parameters, the network introduces a separate output layer for each parameter type. The layer performs regression and outputs normalized values in the range $[0, 1]$. For boolean parameters, values $1$ and $0$ correspond to `true` and `false`, respectively. For integer parameters, the output of the network is mapped and discretized into the range of allowed values.

**Model training**    At a high level, our training is based on the `DAgger` method [40] and is shown in Algorithm 1. The training starts with a randomly initialized policy used to sample the top $K$ most likely strategies for each formula in the training dataset $\mathcal{F}$ (line 4). Selected strategies are evaluated and used to create a training dataset (line 5, described below) on which the policy $\pi$ is retrained (line 6). As the model is initially untrained the strategies are sampled at random and only as the training progresses the policy will learn to select strategies that perform well on formulas from $\mathcal{F}$. As an alternative, one could pre-train the initial policy using strategies supplied by an expert in which case the algorithm would correspond to imitation learning. However, in our work we assume such expert strategies are not available and therefore we start with a model that is initialized randomly.

**Building the training dataset**    Each sample in our training dataset $\mathcal{D} = \{\langle(\boldsymbol{t}_i, \boldsymbol{p}_i), s_i\rangle\}_{i=1}^{M}$ for the neural network consists of a state and its associated training label. Here, the label is a tuple consisting of a probability distribution over tactics $\boldsymbol{t} \in \mathbb{R}^{|\texttt{Tactics}|}$ and the values of all tactic parameters $\boldsymbol{p} \in \mathbb{R}^{|\texttt{Param}|}$. The intuition behind this choice is that for a state $s$, the vector $\boldsymbol{t}$ encodes the likelihood that a given tactic is fast at solving the formula whereas $\boldsymbol{p}$ contains the best parameter values found so far during training. Generating the dataset in this way encodes the preference for strategies that are most efficient in contrast to finding any strategy that solves the input formula. To train the neural network model using such a dataset, the loss is constructed as a weighted average of the cross-entropy loss for tactic prediction and mean-squared-error for parameter prediction.

We build the dataset as follows. First, we evaluate each strategy in $\mathcal{Q}$ on the formula for which it was generated and keep only those that succeeded in deciding the formula. Second, let us denote with $r(t, s_i)$ the best runtime (or `timeout`) achieved from state $s_i$ by first applying tactic $t$, with $r_{best}(s_i)$ the best runtime achieved from state $s_i$, and with $v(p, s_i)$ the value assigned to parameter $p$ in tactics with the best runtime from state $s_i$. We obtain $r$, $r_{best}$ and $v$ by considering all the states and actions performed when evaluating the strategies in $\mathcal{Q}$. Then, for each non-terminal state $s_i$ that eventually succeeded we create one training sample $\langle(\sigma(\tilde{\boldsymbol{t}}_i), \boldsymbol{p}_i), s_i\rangle$ where $\tilde{\boldsymbol{t}}_i = [1/r(t, s_i)]_{t \in \texttt{Tactics}}$, $\sigma$ normalizes $\tilde{\boldsymbol{t}}_i$ to a valid probability distribution and $\boldsymbol{p}_i = [v(p, s_i)]_{p \in \texttt{Param}}$. To generate the training dataset for bilinear model we follow a similar procedure with the exception that we use $\tilde{\boldsymbol{t}}_i = \mathbb{1}[r_{best}(s_i) = r(t, s_i)]_{t \in \texttt{Tactics}}$ which assigns probability 1 to the best tactic and 0 to others.

**Pruning via equivalence classes**    A challenge in training the models presented above is that whether a strategy solves the formula is known only at terminal states. This is especially problematic for datasets where majority of the effort is spent on finding any successful strategy. To address this issue we take advantage of the fact that some information can be learned also from partial strategies – namely their current runtime and their transformed formula. This allows us to check if multiple transformed formulas are equivalent (we consider two formulas equivalent if their abstract syntax tress are identical) and keep only the one which was fastest to reach (and prune the others).

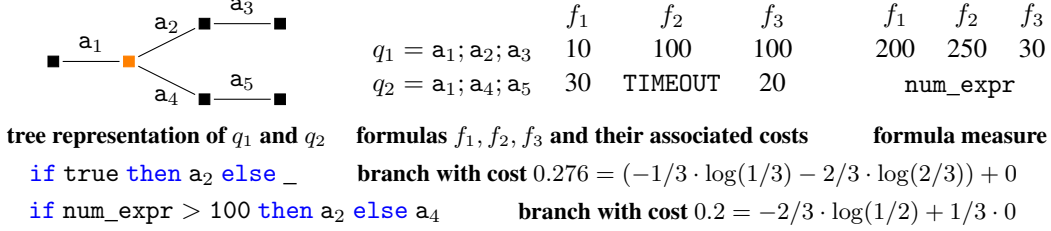

| | $f_1$ | $f_2$ | $f_3$ | $f_1$ | $f_2$ | $f_3$ |
|---|---|---|---|---|---|---|
| $q_1 = \mathtt{a_1; a_2; a_3}$ | 10 | 100 | 100 | 200 | 250 | 30 |
| $q_2 = \mathtt{a_1; a_4; a_5}$ | 30 | TIMEOUT | 20 | | num_expr | |

tree representation of $q_1$ and $q_2$    **formulas $f_1, f_2, f_3$ and their associated costs**    **formula measure**

`if true then a₂ else _`    **branch with cost** $0.276 = (-1/3 \cdot \log(1/3) - 2/3 \cdot \log(2/3)) + 0$

`if num_expr > 100 then a₂ else a₄`    **branch with cost** $0.2 = -2/3 \cdot \log(1/2) + 1/3 \cdot 0$

Figure 2: Illustration of decision tree representation of strategies, scored formulas and branches.

## 4.2 Synthesizing a combined, interpretable strategy

The policy $\pi$ learned in Section 4.1 can be used to speed-up SMT solving as follows: invoke the solver with the current formula $f$ and the action $a_0$ selected by $\pi$, obtain a new intermediate formula $f_1$ from the solver, then again invoke the solver from scratch with $f_1$ and a new action $a_1$, obtaining another intermediate formula $f_2$, and so on. Unfortunately, even if the final formula $f_k$ is solved (e.g., determined to be SAT), we would have lost the connection with the original formula $f$ as at each step we are making a new, fresh invocation of the SMT solver. This is problematic for tasks (e.g., planning) that require more information from SMT solvers, beyond satisfiability of $f$, for instance, the model itself. To address this challenge without changing the internals of the SMT solver, we propose to synthesize an interpretable policy $q_{best}$ that follows $\pi$ and can be directly expressed in the `Strategy` language from Fig. 1 (and thus be given as input to the SMT solver).

Recall from Fig. 1 that the `Strategy` language defines two types of statements that can be used to combine programs: `if-then-else` and `or-else`. However, notice that the `or-else` statement is in fact a special version of the `if` statement with a condition that checks if a given tactic terminates within `c` milliseconds. As a result, we can reduce the problem of synthesizing programs in the `Strategy` language to the problem of synthesizing branches with predicates over a set of candidate strategies. We can obtain the set of candidate strategies by either evaluating $\pi$ over the training formulas or by extracting successful strategies from the set $\mathcal{Q}$ explored during policy learning. Next, we discuss how to synthesize $q_{best}$.

**Decision tree learning** To synthesize branches with predicates, we adapt techniques based on decision tree learning. Consider the tree illustrated in Fig. 2 (top left) with the same structure as used during search (i.e., edges denoting actions and nodes denoting states) but formed from the two candidate strategies $q_1$ and $q_2$. Each SMT formula is evaluated by taking a path in the tree and applying all the corresponding actions. Intuitively, at each node with more than one outgoing edge a decision needs to be taken to determine which path to take. To encode such decisions, our goal is to introduce one or more branches at each such node (denoted as orange square in Fig. 2).

More formally, let $\mathcal{Q} = \{q_1, \dots, q_n\}$ be a set of candidate strategies, $\mathcal{F}$ be our dataset of formulas and $b ::= \mathtt{if\ p\ then}\ q_{\mathtt{true}}\ \mathtt{else}\ q_{\mathtt{false}}$ a branch that partitions $\mathcal{F}$ into two parts – $\mathcal{F}_{\mathtt{true}}$ are formulas on which predicate p evaluates to true and $\mathcal{F}_{\mathtt{false}}$ for the rest. We define a notion of multi-label entropy of a dataset of formulas, denoted as $H(\mathcal{F})$ [14]:

$$H(\mathcal{F}) = -\sum\nolimits_{q \in \mathcal{Q}} p(q) \log(p(q)) + (1 - p(q)) \log(1 - p(q))$$

where $p(q)$ denotes the ratio of formulas solved by strategy $q$ in $\mathcal{F}$. The goal of synthesis is then to discover branches that partition $\mathcal{F}$ into smaller sets, each of which has small entropy (i.e. either none or all formulas are solved). Using the entropy, we define a cost associated with a branch $b$ as:

$$cost(b, \mathcal{F}_{\mathtt{true}}, \mathcal{F}_{\mathtt{false}}) = (|\mathcal{F}_{\mathtt{true}}|/|\mathcal{F}|)H(\mathcal{F}_{\mathtt{true}}) + (|\mathcal{F}_{\mathtt{false}}|/|\mathcal{F}|)H(\mathcal{F}_{\mathtt{false}})$$

That is, branch cost is a weighted sum of entropies in each of the resulting branches. With this scoring function we build the decision tree in a top-down fashion – for each node with multiple outgoing edges we recursively synthesize predicates that minimize the *cost*. If dataset size for some node is below a certain threshold we greedily select the strategy which can solve the most formulas, breaking ties using runtimes. To express branches, we consider the following types of predicates: (i) `true` which allows expressing a default choice, (ii) `Probes` with arithmetic expression as defined in Fig. 1, and (iii) `try s for c` which allows checking whether tactics terminate within `c` ms.

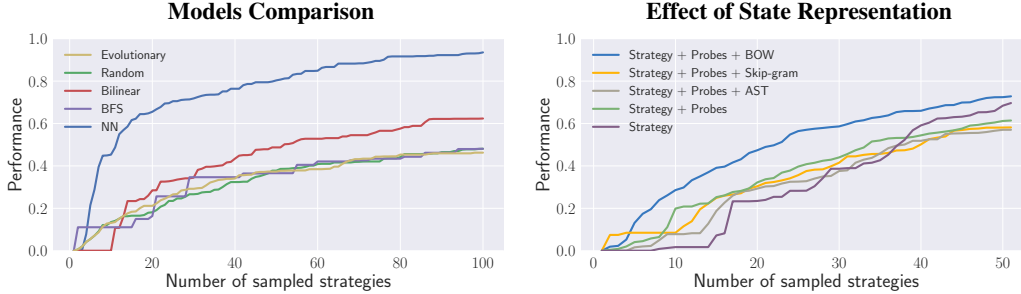

Figure 3: Comparison of proposed models and baselines for learning SMT solver strategies. Each line (higher is better) denotes the quality of the best learned strategy among top $x$ most likely strategies found by a given model (x axis) proportional to the best known strategy overall (y axis).

## 5  Experiments

We implemented our method in a system called `fastSMT` and evaluated its effectiveness on 5 different datasets – `AProVE` [19, 5], `hycomp` [7], `core` [6], `leipzig` [8] and `Sage2` [9, 21]. These contain formulas of varying complexity across 3 logics `QF_NRA`, `QF_BV` and `QF_NIA`. The formulas have on average 336, 35, 228, 153, 345 assertions, 929, 10690, 1672, 886, 1887 expressions and 118, 49, 46, 20, 79 variables for `leipzig`, `core`, `hycomp`, `AProVE` and `Sage2` benchmarks, respectively and require up to 60 MB for each formula to be stored in the SMT2-lib format. All datasets are part of the official `SMT-LIB` benchmarks [10] and encode formulas from both academic and industrial tools (i.e. `Sage` and `AProVE`). For all datasets we split formulas in training, validation and test set in predetermined ratios. To train our models we use 10 iterations of Algorithm 1 with exponentially decaying exploration rate to choose between policy and random action. We train the bilinear model using `FastText` [26] for 5 epochs with learning rate 0.1 and 20 hidden units. We implemented the neural network model in `PyTorch` [37] and trained using a learning rate $10^{-4}$, mini batch size 32, Adam optimizer [31] with Xavier initialization [20] and early stopping. All our datasets, experiments, implementation and extended version of this paper are available at `http://fastsmt.ethz.ch/`.

**Search models comparison**    To compare different learned models we used them to obtain the 100 most likely strategies for each formula in our test dataset. The results are shown in Fig. 3 (left) and additionally include three baseline models that perform random search, breadth-first search and search using an evolutionary algorithm. The x-axis shows the number of most likely strategies sampled from each model and y-axis their runtime proportional to runtime of the best known strategy (i.e., best strategy from top 100 strategies across all models). Here, score one denotes that the best known strategy was found whereas score zero denotes that no strategy was found that solves the formula. Even though baselines perform poorly, they are already able to find simpler strategies that can solve some of the formulas. Note that in our experiment the evolutionary algorithm performed similarly to a random model as it got easily stuck in local minima without enough exploration. Overall, the best model is the neural network which is also most complex and considers the richest set of features.

**Effect of state representation**    In Fig. 3 (right) we evaluate the effect of instantiating the neural network model with a different set of input features capturing the current state. For our task, the representation at the right level of complexity is bag-of-words – if the formula representation is flattened using pre-trained embeddings it loses the relevant information and with more complex AST features the model suffers due to data sparsity. Further, we note that for our task the most important features are those capturing sequences of tactics applied so far, which is illustrated by the strong performance of `Strategy` only model in Fig. 3 (right).

**Comparison to state-of-the-art SMT solvers**    We evaluate the effectiveness of the learned strategies compared to the hand-crafted strategies in Z3 4.6.2 on two metrics – number of solved formulas and runtime. Unless stated otherwise, we use time limit of 10 seconds allocated for solving each formula and instead of runtime, we use the amount of executed basic operations (using `rlimit` counter in Z3) as a deterministic and machine independent measure of the work required to solve the formula. We include additional experiments with runtime in the extended version of the paper.

Table 2: Comparison of best strategy found by any of our models (Section 4.1) against Z3.

| | Formulas solved | | | | Speedup percentile | | |
|---|---|---|---|---|---|---|---|
| | Both | Only Z3 | Only Learned | None | 90th | 50th | 10th |
| `leipzig` | 57 | 3 | 0 | 8 | $5.8\times$ | $60.7\times$ | $191.5\times$ |
| `Sage2` | 332 | 2 | 246 | 220 | $1.2\times$ | $2.7\times$ | $35.5\times$ |
| `core` | 270 | 0 | 0 | 0 | $1.2\times$ | $1.3\times$ | $1.9\times$ |
| `hycomp` | 273 | 0 | 34 | 18 | $1.0\times$ | $1.8\times$ | $4.0\times$ |
| `AProVE` | 283 | 0 | 18 | 14 | $3.9\times$ | $87.8\times$ | $1314.0\times$ |
| Total | 68.3% | 0.3% | 16.8% | 14.6% | $2.6\times$ | $30.9\times$ | $309.4\times$ |

Table 3: Comparison of the combined strategy synthesized by our approach (Section 4.2) against Z3.

| | Formulas solved | | | | Speedup percentile | | |
|---|---|---|---|---|---|---|---|
| | Both | Only Z3 | Only Learned | None | 90th | 50th | 10th |
| `leipzig` | 55 | 5 | 1 | 7 | $1.4\times$ | $9.9\times$ | $21.7\times$ |
| `Sage2` | 2488 | 200 | 705 | 3051 | $0.8\times$ | $1.2\times$ | $3.1\times$ |
| `core` | 270 | 0 | 0 | 0 | $1.2\times$ | $1.3\times$ | $1.8\times$ |
| `hycomp` | 1547 | 93 | 112 | 230 | $0.4\times$ | $1.1\times$ | $2.3\times$ |
| `AProVE` | 1365 | 76 | 112 | 159 | $3.2\times$ | $65.1\times$ | $860.8\times$ |
| Total | 54.6% | 3.6% | 8.9% | 32.9% | $1.4\times$ | $15.7\times$ | $178.0\times$ |

Table 2 shows the number of formulas solved by Z3 compared to the best strategy found by any of our methods. We also measure speedups of our strategies over Z3 on all formulas which were solved by both methods. For example, the 90th percentile in the AProVE benchmark denotes that for 90% of formulas the speedup is at least $3.9\times$. The learned strategies significantly outperform Z3 across all benchmarks – solving 17% more formulas, often with up to 3 orders of magnitude speedups and with only 5 formulas not solved by any of the learned strategies even though they can be solved by Z3. This shows that, for all our benchmarks, the strategies found during training generalize well to other formulas from the same dataset.

Table 3 shows performance of the single combined strategy synthesized as described in Section 4.2. Here, the result of synthesis is a program in the `Strategy` language that is used as input to Z3 together with the formula to solve. Naturally, as the `Strategy` language has limited expressiveness (i.e., restricting the kind of expressions that can be used as branch predicates) the performance improvement is smaller compared to best strategy found by any of our methods for each formula as shown in Table 2. However, more importantly, the improvement over the default Z3 strategy is still significant allowing us to solve 8.9% more formulas compared to Z3.

**Generalization to harder to solve formulas**    So far, in all our experiments the time limit for both training and evaluation was set to 10 seconds. To evaluate how our learned strategies generalize to harder to solve formulas we kept the 10 seconds time limit for training but used 600 seconds time limit for the evaluation. Then, our learned strategies can solve 97.7% formulas (up from 85.1%) across all the benchmarks with even slightly better speedups than those shown in Table 2.

## 6   Conclusion

We presented a new approach for solving SMT formulas based on a combination of training a policy that learns to discover strategies that solve formulas efficiently and a synthesizer that produces interpretable strategies based on this model. The synthesized strategies are represented as programs with branches and are directly usable by state-of-the-art SMT solvers to guide its search. This avoids the need to evaluate the learned policy at inference time and enables close integration with existing SMT solvers. Our technique is practically effective – it solves 17% more formulas over a range of benchmarks and achieves up to $100\times$ runtime improvements over state-of-the-art SMT solver.

## Footnotes

[1]For more information and examples we refer the reader to the online tutorial available at: `https://rise4fun.com/z3/tutorial/strategies`

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
