[Supplementary Material · supplementary.pdf]

# Learning to Solve SMT Formulas
# Supplementary Material

**Mislav Balunović, Pavol Bielik, Martin Vechev**
Department of Computer Science
ETH Zürich
Switzerland
`bmislav@ethz.ch`, `{pavol.bielik, martin.vechev}@inf.ethz.ch`

We provide two appendices. Appendix 1 includes in depth descriptions of the algorithms used for learning and synthesis. Appendix 2 provides two additional experiments - evaluating with time limit of 10 minutes and the effect of iterative training used by Algorithm 1.

# 1 Additional details on learning and synthesis

**Finding most likely strategies**  As described in Algorithm 1, a key step of training is a procedure that finds the top $K$ most likely strategies to solve a given formula (line 4 of Algorithm 1). This algorithm, shown in Algorithm 2, takes as input a single formula $f \in \mathcal{F}$, a model $\pi$ (e.g., a neural network policy, bilinear model, etc.) and integer $K$ (number of strategies to explore).

During the search we keep a priority queue of tuples $(s_j, a_j, p_j)$ consisting of a state, possible action and its associated probability, initialized with $(s_0, \epsilon, 1)$, $s_0$ denoting initial state. At each step, we remove the tuple with highest probability from the queue, apply its action to obtain a new state $s'_j$ and update the priority queue with new tuples $(s'_j, a, p_j \cdot \pi(a \mid s'_j))$ for all actions $a \in \texttt{Tactics} \times \texttt{Params}$ capturing the possible transitions from $s'_j$. For practical reasons we approximate the set of all possible actions (denoted as $\text{ACTIONS}(s'_j, \pi)$) as follows:

- If we are using the neural network policy, we consider the most likely parameters for each tactic according to that policy, or

- If we are not using the neural network policy and instead are using models which do not predict parameters, we consider 20 different parameter configurations for each tactic which are selected at random before training starts.

As described in Section 4.1, we additionally perform pruning of states (line 9) which can not possibly lead to an optimal strategy. Finally, we note that in practice we perform the search for batch of formulas at once in order to leverage parallelization capabilities of our system.

---

**Algorithm 2:** Procedure for finding top K strategies

**Data:** Formula $f$, Model $\pi$, Number of strategies to sample $K$
**Result:** Top $K$ most likely strategies according to the model

1   $s_0 = (f, \epsilon)$
2   $S = \emptyset$
3   $queue = \text{PRIORITY\_QUEUE}()$
4   $queue.\text{PUSH}(s_0, \epsilon, 1)$
5   **for** $j = 1 : K$ **do**
6      $(s_j, a_j, p_j) = queue.\text{POP}()$
7      $s_j \xrightarrow{a_j} s'_j$          ▷ State $s'_j$ is obtained by applying action $a_j$ in state $s_j$
8      $S \leftarrow S \cup \{s'_j.strategy\}$     ▷ Strategy is extracted from state $s'_j$ and added to $S$
9      **if** $\neg$ PRUNED$(s'_j)$ **then**
10         **for** $a \in \text{ACTIONS}(s'_j, \pi)$ **do**
11            $queue.\text{PUSH}(s'_j, a, p_j \cdot \pi(a \mid s'_j))$

12   **return** $S$

---

**Building the training dataset**  In Section 4.1 we described how we construct the dataset used to train the neural network policy. Here we illustrate the process on a concrete example. Let us consider an example where the model explored seven different states as shown in Fig. 4 (left). Further, in addition to the visited states we also keep information about the cumulative runtime required to compute the state (starting from the initial state $s_0$) as well as whether the state successfully solves the formula. Then, according to the procedure described in Section 4.1, the dataset is constructed by generating one training sample $\langle (\sigma(\boldsymbol{t}_i), \boldsymbol{p}_i), s_i \rangle$ for each non-terminal state that eventually succeeded in solving the formula. In our example, this corresponds to generating one training sample for states $s_1$, $s_2$ and $s_3$ as shown in Table 4 where we use $\epsilon$ to denote that no tactic was applied so far. Note that for state $s_1$ there are two possible tactics which lead to solving the formula ($\texttt{simplify(flat} = \texttt{true)}$ and $\texttt{bit\_blast}$) and therefore we assign non-zero target probability to both of these tactics. However, the best strategies in respective subtrees have different runtimes hence probabilities assigned to the corresponding tactics are different as shown in Table 4. Finally, in case both states $s_6$ and $s_7$ would not solve the formula, it would mean that no training example is generated for state $s_3$ (since we generate training samples only for non-terminal states that eventually succeeded in solving the formula).

$$r(s_4) = 10$$
$$r(s_5) = \text{TIMEOUT}$$
$$r(s_6) = \text{TIMEOUT}$$
$$r(s_7) = 40$$

Figure 4: Example with all states visited during solving a formula. Terminal states are colored blue if formula was solved and red otherwise. Runtime for each terminal state is shown on the right.

| State | Target tactics | Target parameters |
|---|---|---|
| $s_1 = (\varphi_1, \epsilon)$ | $\Pr(\texttt{simplify}) = 0.8$ <br> $\Pr(\texttt{bit\_blast}) = 0.2$ | $\texttt{flat} = \texttt{true}$ <br> - |
| $s_2 = (\varphi_2, \texttt{simplify}(\texttt{flat} = \texttt{true}))$ | $\Pr(\texttt{solve\_eqs}) = 1$ | - |
| $s_3 = (\varphi_3, \texttt{bit\_blast})$ | $\Pr(\texttt{sat}) = 1$ | $\texttt{scc} = \texttt{false}$ |

Table 4: Dataset constructed from the example shown in Fig. 4.

**Reducing the set of strategies**   As the set of synthesized strategies can be large we perform synthesis (described in Section 4.2) only on a subset set of strategies. Intuitively, these strategies should be: strong individually (i.e. each strategy should be able to solve large number of formulas alone) and strong together (i.e. number of formulas solvable by at least one of the strategies should be large). In order to trade-off these two conditions we use greedy procedure shown in Algorithm 3.

We proceed in an iterative manner, choosing one new strategy at every step. In every iteration, strategy receives a score for every formula that it can solve. Score is equal to $\lambda$ for every formula which was previously unsolved, and $1 - \lambda$ if formula was already solved (by another previously selected strategy). One can notice that if $\lambda = 1$ algorithm will try to greedily maximize the number of formulas that strategies can solve in the union. If $\lambda = 1/2$ algorithm will simply select $k$ strategies which can solve the most formulas individually. In our experiments we treat $\lambda$ as a hyperparameter and optimize it on a validation set of formulas. Concretely, we set $\lambda$ to 0.5, 0.99, 0.75, 0.95, 0.7 for `Sage2`, `AProVE`, `leipzig`, `core` and `hycomp` respectively.

---

**Algorithm 3:** Greedy strategies selection

---

**Data:** Set of formulas $\mathcal{F}$, Strategies $s_1, ..., s_m$, weight $\lambda \in [0, 1]$
**Result:** Set $S$ consisting of $k$ selected strategies
Initialize $S_{best} = \emptyset$
Initialize $A = \mathcal{F}, B = \emptyset$
**for** $iter = 1 : k$ **do**
    **for** $i = 1 : m$ **do**
        **if** $s_i \notin S_{best}$ **then**
            Calculate $A_i$ = subset of formulas in $A$ which strategy $s_i$ can solve
            Calculate $B_i$ = subset of formulas in $B$ which strategy $s_i$ can solve
            Define $score_i = \lambda|A_i| + (1 - \lambda)|B_i|$
    Add strategy $s_i$ with highest corresponding $score_i$ to $S_{best}$
    $A = A \setminus A_i$
    $B = B \cup A_i$

---

## 2   Additional experiments

**Evaluation with 10 minute time limit**    The 10 second time limit in our experiments was selected for practical purposes - it is large enough to solve 83.3% of the formulas and to learn strategies in a reasonable time. To check how well our strategies generalize to higher time limits we kept the 10 seconds time limit for training, but used 10 minutes for evaluation. We show results from this experiment in Table 6. With the 10 minute time limit, 88.1% of formulas are solved by both methods. Crucially, our strategies are still able to solve 8.6% more formulas than Z3. In addition, the speedups over Z3 are comparable (and even slightly higher) to speedups achieved with 10 second time limit. For comparison, Table 2 shows the results of evaluating the same strategies on the same set of formulas using a 10 second time limit. Note since the experiments take significantly longer to run we evaluated them only on a subset of the dataset.

**Evaluation on larger set of formulas**    For completeness, we also include the results of the best strategies synthesized by our models on the full set of formulas in our test set using the 10 second time limit. Results are shown in Table 5. The set of formulas is the same as in Table 3 (as opposed to Table 2 and Table 6 which use a subset of the formulas).

**Number of basic operations and runtime**    As stated in Section 4, we use the number of basic operations as a deterministic measure of the amount of work required to solve a formula. In Table 7, we show a comparison between the number of operations and wall clock time for the experiments in Table 6. Note that wall clock time is often imprecise. Especially for formulas which can be solved very fast, wall clock time mostly accounts for initialization of the solver and overhead of the system.

**Effect of iterative training**    In Fig. 5 we show the improvement of our neural network policy as it is continuously retrained using DAgger. We perform a total of 10 stages of DAgger, retraining the model after every stage. In every stage, the current model is used to search for the best strategies, as described in Algorithm 2.

For the purpose of this experiment, we save the models after 2, 4, 6, 8 and 10 stages of DAgger. Then we load each model again and use it to search for the best strategies on the unseen formulas from the test set. We run all of the models for 100 iterations without retraining (which means that each model predicts 100 best strategies for every formula).

One can notice that later models tend to outperform earlier models, thus justifying the increased number of training stages. The only exception in this case are models trained after 6 and 8 stages where an earlier model performs better by a small margin. This can be explained by the stochastic nature of the training procedure.

**Tactics**    For completeness, we also include the set of tactics and parameters used in our experiments. Tactics used for `Sage2` and `core`, both in QF_BV logic are shown in Table 8. Tactics for `hycomp` in QF_NRA logic are in Table 9. Finally, tactics for `leipzig` and `AProVE` in logic QF_NIA are in Table 10. For every tactic we list the parameters that we used (all parameters not listed here are set to the default value) as well as their types.

Table 5: Comparison of best strategy found by any of our models (Section 4.1) against Z3.

| | Formulas solved | | | | Speedup percentile | | |
|---|---|---|---|---|---|---|---|
| | Both | Only Z3 | Only Learned | None | 90th | 50th | 10th |
| leipzig | 57 | 0 | 3 | 8 | 5.8× | 60.7× | 191.5× |
| Sage2 | 2531 | 0 | 2402 | 1511 | 1.2× | 2.5× | 22.0× |
| core | 270 | 0 | 0 | 0 | 1.2× | 1.3× | 1.9× |
| hycomp | 1633 | 1 | 210 | 138 | 1.0× | 2.0× | 4.3× |
| AProVE | 1397 | 3 | 221 | 91 | 4.0× | 89.6× | 988.7× |
| Total | 56.2% | 0.1% | 27.0% | 16.7% | 2.6× | 31.2× | 241.7× |

Table 6: Comparison of best strategy found by any of our models against Z3 with 10min time limit.

| | Formulas solved | | | | Speedup percentile | | |
|---|---|---|---|---|---|---|---|
| | Both | Only Z3 | Only Learned | None | 10th | 50th | 90th |
| leipzig | 63 | 0 | 1 | 4 | 3.5× | 43.9× | 183.2× |
| Sage2 | 630 | 0 | 138 | 32 | 1.3× | 6.5× | 199.6× |
| core | 270 | 0 | 0 | 0 | 1.2× | 1.3× | 1.9× |
| hycomp | 298 | 0 | 10 | 17 | 1.0× | 2.0× | 40.1× |
| AProVE | 306 | 0 | 3 | 6 | 3.9× | 89.3× | 1301.5× |
| Total | 88.1% | 0.00% | 8.6% | 3.3% | 2.2× | 28.6× | 345.3× |

Table 7: Comparison of speedup in number of operations and wall clock time.

| | Speedup - number of operations | | | | Speedup - Wall clock time | | | |
|---|---|---|---|---|---|---|---|---|
| | $P_{10}$ | $P_{50}$ | $P_{90}$ | Mean | $P_{10}$ | $P_{50}$ | $P_{90}$ | Mean |
| leipzig | 3.5× | 43.9× | 183.2× | 71.6× | 0.3× | 2.3× | 7.3× | 3.5× |
| Sage2 | 1.3× | 6.5× | 199.6× | 62.7× | 1.2× | 4.8× | 72.5× | 37.8× |
| core | 1.2× | 1.3× | 1.9× | 1.4× | 0.5× | 0.8× | 1.3× | 0.9× |
| hycomp | 1.0× | 2.0× | 40.1× | 51.9× | 0.9× | 1.4× | 65.7× | 25.0× |
| AProVE | 3.9× | 89.3× | 1301.5× | 519.9× | 0.9× | 6.4× | 120.8× | 45.8× |
| Total | 2.2× | 28.6× | 345.3× | 141.5× | 0.8× | 3.1× | 53.5× | 22.6× |

Figure 5: Performance of neural network after 2, 4, 6, 8 and 10 retraining iterations.

Table 8: Tactics and parameters used for QF_BV logic (`Sage2` and `Core` benchmarks).

| Tactics | Parameters | Type |
|---|---|---|
| | elim_and | bool |
| simplify | blast_distinct | bool |
| | push_ite_bv | bool |
| | som | bool |
| | pull_cheap_ite | bool |
| | hoist_mul | bool |
| | local_ctx | bool |
| | flat | bool |
| smt | - | - |
| bit_blast | - | - |
| bv1_blast | - | - |
| solve_eqs | - | - |
| aig | aig_per_assertion | bool |
| qfnra_nlsat | - | - |
| sat | - | - |
| max_bv_sharing | - | - |
| reduce_bv_size | - | - |
| purify_arith | - | - |
| propagate_values | push_ite_bv | bool |
| elim_uncnstr | - | - |
| ackermannize_bv | - | - |

Table 9: Tactics and parameters used for QF_NRA logic (`hycomp` benchmark).

| Tactics | Parameters | Type |
|---|---|---|
| | elim_and | bool |
| simplify | blast_distinct | bool |
| | som | bool |
| | hi_div0 | bool |
| | hoist_mul | bool |
| | local_ctx | bool |
| | flat | bool |
| smt | - | - |
| bit_blast | - | - |
| solve_eqs | - | - |
| | - | - |
| qfnra_nlsat | inline_vars | bool |
| | factor | bool |
| | seed | int |
| max_bv_sharing | - | - |
| propagate_values | push_ite_bv | bool |
| elim_uncnstr | - | - |
| nla2bv | nla2bv_max_bv_size | int |
| ctx_simplify | - | - |

Table 10: Tactics and parameters used for QF_NIA logic (`leipzig` and `AProVE` benchmarks).

| Tactics | Parameters | Type |
|---|---|---|
| | elim_and | bool |
| simplify | som | bool |
| | blast_distinct | bool |
| | flat | bool |
| | hi_div0 | bool |
| | local_ctx | bool |
| | hoist_mul | bool |
| propagate_values | push_ite_bv | bool |
| smt | - | - |
| bit_blast | - | - |
| solve_eqs | - | - |
| qfnra_nlsat | - | - |
| max_bv_sharing | - | - |
| elim_uncnstr | - | - |
| nla2bv | nla2bv_max_bv_size | int |
| ctx_simplify | - | - |