[Reviews · NeurIPS 2018]

Reviewer 1



The paper discusses the problem of solving SMT formula by means of combinations of operations on formulas called tacticts. Such combinations form simple programs called strategies operating on a state represented by the formula. The paper presents an approach for learning the strategies. The approach works in two phases: first a policy is learned, that defines a probability distribution over tacticts given the state, then this is transformed into a set of sequences of tactics that are combined into strategies by adding branching instructions. The approach is experimentally evaualated by comparing its performance with respect to hand crafted strategies using the Z3 SMT solver. The learned strategies are able to solve 20% more problems within the timeout and achieve speedups of x1000 in some cases. The idea of applying machine learning to induce SMT strategies is very promising, with several works pursuing it. The paper is clearly distinguished from previous work for its focus on strategies, simple program combining elementary operations. The only other work performing strategy learning uses genetic algorithms. The experimental results are impressive and show that the approach outperforms evolutionary approaches that get stuck in local minima. However, the presentation left some points not sufficiently clarified. In particular, you say that you first induce policies that are probability distribution over tactics given the STATE, which means the formula. In other points it seems that policies are probability distribution over sequences of tacticts, please clarify. How do you convert policies to sequences of tactics? Do you replay the policies over the training formulas? What is the difference with using your procedure for generating the training set to generate the best sequence of tactics for each formula and use those as input for the second phase? In your second phase you do not consider the repeat instruction to build the strategies, why? The paper also contains notation imprecisions: the log-likelihood formula for the bilinear model is incorrect, as y_i is Boolean and the output of \sigma(UV...) is a vector, please correct, I guess you want to use cross-entropy. At the end of Section 4 you say that "try s for c" is a predicate, while it is a strategy (Table 1). The difference between the combination of tactics with ; or with or-else is not clear: on page 8 you say that for the strategy “simplify(local_ctx : true); sat; bit_blast; sat“. " either the first one already solves the formula or it fails in which case bit_blast; sat is executed" so ; and or-else seem the same. Please clarify what are the speedup percentiles in Table 2

Reviewer 2



Thank you for the detailed rebuttal! I have accordingly updated my score from 6 to 7. ---------------------------- This paper proposes a framework for learning solving strategies for Satisfiability Modulo Theories (SMTs). SMT generalizes SAT, and as such can be used to model a variety of verification applications, among others. Given an instance, an SMT solver such as Z3 will select a strategy, i.e. a sequence of transformations, that will be applied step by step to the input formula, culminating in solving it and returning the answer (true or false). The choice of strategies in SMT solvers is currently heuristic or manually optimized for a set of instances. Alternatively, this paper proposes to learn strategies based on training instances and label-strategies resulting from exploration. A set of learned strategies are then synthetized into a decision tree structure than can be easily fed into the Z3 solver. Experimentally, it is shown that the proposed approach can boost the performance of Z3 significantly. I am rather positive about this paper, hence my score of 6/10. The paper is relatively well-written, although some points could be clarified here and there. The problem this paper addresses is definitely interesting, and part of a recent effort that aims at improving constraint solvers for MIP/SAT/CSP/SMT using machine learning. I really appreciate the 2-step methodology proposed by the authors, as they do not try to ignore the advances in SMT solvers, but rather hybridize Z3 with their learned strategies. I think that the main weakness of the paper, in its current version, is the experimental results. This is not to say that the proposed method is not promising - it definitely is. However, I have some questions that I hope the authors can address. - Time limit of 10 seconds: I am quite intrigued as to the particular choice of time limit, which seems really small. In comparison, when I look at the SMT Competition of 2017, specifically the QF_NIA division (http://smtcomp.sourceforge.net/2017/results-QF_NIA.shtml?v=1500632282), I find that all 5 solvers listed require 300-700 seconds. The same can be said about QF_BF and QF_NRA (links to results here http://smtcomp.sourceforge.net/2017/results-toc.shtml). While the learned model definitely improves over Z3 under the time limit of 10 seconds, the discrepancy with the competition results on similar formula types is intriguing. Can you please clarify? I should note that while researching this point, I found that the SMT Competition of 2018 will have a "10 Second wonder" category (http://smtcomp.sourceforge.net/2018/rules18.pdf). - Pruning via equivalence classes: I could not understand what is the partial "current cost" you mention here. Thanks for clarifying. - Figure 3: please annotate the axes!! - Bilinear model: is the label y_i in {-1,+1}? - Dataset statistics: please provide statistics for each of the datasets: number of formulas, sizes of the formulas, etc. - Search models comparison 5.1: what does 100 steps here mean? Is it 100 sampled strategies? - Missing references: the references below are relevant to your topic, especially [a]. Please discuss connections with [a], which uses supervised learning in QBF solving, where QBF generalizes SMT, in my understanding. [a] Samulowitz, Horst, and Roland Memisevic. "Learning to solve QBF." AAAI. Vol. 7. 2007. [b] Khalil, Elias Boutros, et al. "Learning to Branch in Mixed Integer Programming." AAAI. 2016. Minor typos: - Line 283: looses -> loses

Reviewer 3



This paper studies machine learning techniques to learn heuristics that help guide SMT solvers (in particular Z3) to speed up finding solutions. The problem of finding a strategy for solving an SMT formula is formulated as a reinforcement learning problem where the cost function is (essentially) runtime. The research is well executed and shows taste in the problem setup and choice of solution techniques (in particular, employing a combination of Imitation Learning with a Neural Network policy, MCTS and Decision Trees). This kind of research seems very promising in improving SMT solvers in the future. I'd also love to see a full fletched reinforcement learning algorithm like PPO employed here, which could lead to further improvements. The paper is well written and understandable. Two points to consider: - I very much encourage the authors to publish the code to reproduce their experiments. In fact it would be considered a serious drawback if the code was not released. This is the kind of paper that is very hard to reproduce without code because many details need to be gotten right. Even - In terms of possible improvements, I would like to see a short discussion on the influence of neural network evaluation time on the speedups. If I understand correctly, speed was measured in inference operations per second and thus does not account for the neural network evaluations. How would the picture change if the NN evaluations would be taken into account? If these two points were addressed in a convincing way, I would consider bumping my review score from "6" to "7". After reading the author feedback, I adapt my review score to 7 and hope the paper will be published. Great job!